# Gender-Specific Effects on the Cardiorespiratory System and Neurotoxicity of Intermittent and Permanent Low-Level Lead Exposures

**DOI:** 10.3390/biomedicines12040711

**Published:** 2024-03-22

**Authors:** Liana Shvachiy, Ângela Amaro-Leal, Filipa Machado, Isabel Rocha, Tiago F. Outeiro, Vera Geraldes

**Affiliations:** 1Center for Biostructural Imaging of Neurodegeneration, Department of Experimental Neurodegeneration, University Medical Center Göttingen, 37075 Göttingen, Germany; lianashvachiy@fm.ul.pt (L.S.); tiago.outeiro@med.uni-goettingen.de (T.F.O.); 2Cardiovascular Centre of the University of Lisbon, 1649-028 Lisbon, Portugal; filipadcmachado@gmail.com (F.M.); isabelrocha0@gmail.com (I.R.); 3Institute of Physiology, Faculty of Medicine, University of Lisbon, 1649-028 Lisbon, Portugal; araqueleal@hotmail.com; 4Egas Moniz Center for Interdisciplinary Research (CiiEM), Egas Moniz School of Health & Science, 2829-511 Almada, Portugal; 5Max Planck Institute for Natural Science, 37075 Göttingen, Germany; 6Translational and Clinical Research Institute, Faculty of Medical Sciences, Newcastle University, Newcastle upon Tyne NE2 4HH, UK; 7Scientific Employee with an Honorary Contract at Deutsches Zentrum für Neurodegenerative Erkrankungen (DZNE), 37073 Göttingen, Germany

**Keywords:** lead exposure, gender differences, hypertension, reactive astrogliosis

## Abstract

Lead exposure is a significant health concern, ranking among the top 10 most harmful substances for humans. There are no safe levels of lead exposure, and it affects multiple body systems, especially the cardiovascular and neurological systems, leading to problems such as hypertension, heart disease, cognitive deficits, and developmental delays, particularly in children. Gender differences are a crucial factor, with women’s reproductive systems being especially vulnerable, resulting in fertility issues, pregnancy complications, miscarriages, and premature births. The globalization of lead exposure presents new challenges in managing this issue. Therefore, understanding the gender-specific implications is essential for developing effective treatments and public health strategies to mitigate the impact of lead-related health problems. This study examined the effects of intermittent and permanent lead exposure on both male and female animals, assessing behaviours like anxiety, locomotor activity, and long-term memory, as well as molecular changes related to astrogliosis. Additionally, physiological and autonomic evaluations were performed, focusing on baro- and chemoreceptor reflexes. The study’s findings revealed that permanent lead exposure has more severe health consequences, including hypertension, anxiety, and reactive astrogliosis, affecting both genders. However, males exhibit greater cognitive, behavioural, and respiratory changes, while females are more susceptible to chemoreflex hypersensitivity. In contrast, intermittent lead exposure leads to hypertension and reactive astrogliosis in both genders. Still, females are more vulnerable to cognitive impairment, increased respiratory frequency, and chemoreflex hypersensitivity, while males show more reactive astrocytes in the hippocampus. Overall, this research emphasizes the importance of not only investigating different types of lead exposure but also considering gender differences in toxicity when addressing this public health concern.

## 1. Introduction

Lead exposure is a significant and serious health problem, ranking in the top 10 primary toxicants that damage human health [1,2,3]. Unlike certain contaminants, there are no acceptable levels of lead exposure, making it critical to address this issue as soon as possible [1,4,5]. Understanding the many origins and pathways of lead exposure is critical for developing effective methods to minimise its negative consequences.

Lead exposure can occur via a variety of routes, the most common of which are environmental and acute/occupational [1,4,5]. Environmental sources include commonplace factors such as polluted water, lead-based paints, soil, and air pollution [6,7,8,9]. Acute or occupational exposure, on the other hand, is more common in some industries where employees are in direct contact with lead-based products [2,10,11,12,13,14].

Lead exposure has far-reaching health consequences, affecting multiple body systems due to its accumulation in the soft tissues [2,5,6,7,8,9,11,15,16,17,18,19], primarily influencing the cardiovascular and neurological systems. Long-term lead exposure has been related to cardiovascular issues such as hypertension and an increased risk of heart disease [20,21,22,23,24]. Furthermore, the neurological repercussions are especially troubling, since lead exposure has been linked to cognitive deficits, developmental delays, and behavioural abnormalities, particularly in children [25,26,27,28,29,30,31,32].

The biological differences between men and women can have a substantial impact on the health effects of lead exposure [33,34,35,36,37]. Notably, the female reproductive system is especially vulnerable to the negative effects of lead [38,39,40,41,42,43]. Elevated lead levels have been linked to decreased fertility, an increased risk of pregnancy problems, and negative birth outcomes such as miscarriages and premature births [1,4,5]. Furthermore, prenatal lead exposure in children has been associated with developmental delays and cognitive impairments [44,45,46]. Gender-specific patterns in neurotoxicity have been discovered, suggesting unique cognitive effects following lead exposure. Females, particularly throughout critical life periods such as menopause, may have more severe cognitive deficits than males [33,43]. Furthermore, gender disparities in sensitivity to neurobehavioral effects have been observed, with men being more susceptible to behavioural disruptions and women being more susceptible to cognitive deficiencies [38,39,40,42].

The impact of oestrogen, a female hormone, appears to modify the impact of lead exposure on multiple physiological systems. Oestrogen may amplify the harmful effects of lead on the cardiovascular system, potentially increasing the risk of hypertension and cardiovascular disease in women [35,36,38,42,43,47]. On the other hand, oestrogen’s neuroprotective characteristics may attenuate lead’s neurotoxic effects in specific brain areas, thereby contributing to the findings [42].

Globalisation has created new issues in the management of lead exposure [21,48]. Importing items from nations with relaxed lead-content restrictions poses a substantial danger. Toys, cosmetics, and consumer gadgets may contain greater quantities of lead, causing risks, particularly to vulnerable groups such as children [8,49,50]. International travel is also a possible cause of lead exposure. Tourists who visit areas with excessive lead levels in the air or contaminated soil inadvertently expose themselves to health concerns. Mobility programmes are also a source of new lead exposure profiles, namely intermittent lead exposure. Furthermore, the transboundary spread of lead pollution caused by globalisation influences places far from the initial sources of contamination.

Understanding the gender implications of lead poisoning is critical for developing therapies and public health measures. Gender-specific vulnerability-based methods can enhance health outcomes and reduce the burden of lead-related disorders.

## 2. Materials and Methods

### 2.1. Animal Model of Long-Term Lead Exposure

The lead exposure models were developed as described previously, considering ingestion to be the most common exposure route [51,52]. Seven-day-pregnant Wistar rats (Charles River Laboratories, Chatillon-sur-Chalaronne, France) were separated into Pb-treated and control groups. The tap drinking water in the Pb-treated group was replaced with a 0.2% (p/v) lead (II) acetate solution (Acros Organics, Antwerp, Belgium) dissolved in deionized water.

The pups, after weaning at 21 days, were divided into 6 groups according to the type of exposure and gender. The 0.2% lead acetate solution was given to the lead-exposed groups: for female (IntPb F, *n* = 17) and male (IntPb M, *n* = 17) intermittent groups—lead exposure until 12 weeks of age, no exposure (tap water) between 12 and 20 weeks, and second exposure from 20 to 28 weeks of age, and for female (PerPb F, *n* = 17) and male (PerPb M, *n* = 16) permanent groups—lead solution in the diet from foetal period until 28 weeks of age. Tap water was given to the age-matched female (Ctrl F, *n* = 16) and male (Ctrl M, *n* = 19) control groups.

To offer a full functional and morphological evaluation at the endpoint of exposure, all animals were exposed to the same experimental methodology. The experimental methodology complied with European and national animal welfare regulations and was authorised by the Academic Medical Centre of Lisbon (CAML), Portugal, with permission number 411/16.

### 2.2. Behavioral Evaluation

Animals were subjected to a battery of standard behavioural tests two weeks before functional evaluation (at 26 weeks of age) to assess (i) open-field test for locomotor and exploratory behaviour [53], (ii) elevated plus maze for anxiety-like behaviour [54], and (iii) novel object recognition test for evaluation of the episodic long-term memory [55]. During the experimental days, animals were brought into the behaviour testing room for at least 1 h before the start of the testing session. All behavioural studies were carried out between the hours of 8 a.m. and 6 p.m. in a quiet room with low illumination, and all animals were subjected to a four-day handling period [56,57] for the researcher and testing room habituation and to reduce biases.

All behaviour equipment was cleaned with 70% ethanol between animals. All tests were recorded with a UV camera (Chacon, Wavre, Belgium), and the resulting videos were analysed with ANY-maze software, V 7.2 (Stoelting Co., Wood Dale, IL, USA).

#### 2.2.1. Elevated plus Maze Test

An elevated plus maze test (EPM) was performed for anxiety-like behaviour evaluation [32,48,54,58]. The apparatus consists of a 50 cm raised maze with four arms (two open arms (50 × 10 cm) perpendicular to two closed arms (50 × 10 × 30 cm height) that create a plus shape). Each animal was placed in the centre of the labyrinth for 5 min to explore freely, with no prior habituation to the maze, and the following ratio (time spent in open or closed arms/total time) × 100 was used to calculate the percentage of time spent in open and closed arms, respectively [32,48,58].

#### 2.2.2. Open-Field Exploration Test

Exploration of a new environment and general locomotion are commonly evaluated using the open-field test (OFT) taking advantage of the curious nature of the rodents [59]. The OFT apparatus is composed of a square black box (67 × 67 × 57 cm in height) that has been “virtually” split into three concentric squares: (1) the periphery zone (near the walls), (2) the intermediate zone, and (3) the centre. The animals were left in the centre of the maze to explore freely for 5 min, which is generally enough time to assess the specified parameters. We computed the total travelled distance and the average velocity of the animals as main parameters of the test from the evaluation of the central point of the animal [32,48,53,60,61].

#### 2.2.3. Novel Object Recognition Test

A novel object recognition test (NOR) with a 24 h retention interval was used to evaluate the episodic long-term memory using the same protocol as described previously using the OFT arena [32,48,55,62]. Briefly, the objects used (clear and brown glass shapes) were randomized and their position in relation to the other objects was changed to use each object as a source of familiarity or novelty as well as the position change within the maze to exclude spatial biases of the animals.

The test consists of three stages: habituation, training, and testing. During habituation (three days), animals were left to explore the apparatus freely for 15 min. On the fourth day, they were exposed to two familiar (F) items, for 5 min. On the fifth day, the animals were exposed to two objects for 5 min, one previously encountered object (F) and one novel object (N) [55]. The testing day was recorded and analysed using ANY-maze^®^ software, V 7.2 employing 3-point analysis (head, torso, and tail of the animal), and only the data from the head point analysis were relevant for object exploration. The amount of time animals spent around each object during the testing stage was used to quantify exploratory behaviour. The number of approaches that involved smelling, rearing towards, or touching the item was tallied. Exploration did not include sitting rearward to the item or passing in front of it without pointing their nose in the object’s direction [55]. The exploration time was measured as follows: ET (%) = (time exploring the object/overall exploring time) × 100.

The novelty index was then computed as follows:(ET% Novel − ET% Familiar)/(ET% Novel + ET% Familiar)(1)

This index has a range of −1 to 1, with negative values representing the lack of discrimination between the new and familiar items (i.e., spending more time exploring the familiar object or equal time exploring both things) and positive values representing more exploration of the novel object [32,48,63].

### 2.3. Metabolic Evaluation

For 24 h before the acute experiment at 28 weeks, rats were kept in metabolic cages to assess their body weight, food and drink consumption, and urine and faeces production.

### 2.4. Functional Evaluation

#### 2.4.1. Physiological and Autonomic Evaluation

After behavioural and metabolic evaluation, animals were anaesthetized with sodium pentobarbital (60 mg/kg, IP). Anaesthesia levels were maintained using a 20% solution (*v*/*v*) of the same anaesthetic after assessing the withdrawal response. A homoeothermic blanket (Harvard Apparatus, Cambourne, UK) was used to keep the rectal temperature stable. To measure tracheal pressure, the trachea was cannulated below the larynx and the respiratory frequency (RF) was calculated from the measured pressure. The femoral artery and vein were cannulated to monitor blood pressure (BP) and inject saline and drugs, respectively. The electrocardiogram (ECG) was assessed using subcutaneous electrodes in three limbs, and the heart rate was calculated using the ECG data (Neurology, Digitimer, Welwyn Garden City, UK).

The right carotid artery was catheterized, and chemoreceptors were stimulated with lobeline (0.2 mL, 25 g/mL, Sigma, St. Louis, MO, USA) [51]. A phenylephrine injection (0.2 mL, 25 g/mL, Sigma, St. Louis, MO, USA) in the femoral vein was used to stimulate the baroreflexes [32,48,51].

Blood lead levels (BLL) were determined from the venous blood using an atomic absorption spectrophotometer (Shimadzu, Model no. AA 7000, Kyoto, Japan).

#### 2.4.2. Data Acquisition and Analysis

Continuous recordings of blood pressure (BP), ECG, heart rate (HR), and respiratory frequency (RF) were performed (PowerLab, ADInstruments, Colorado Springs, CO, USA). These parameters were acquired, amplified, and filtered at 1 kHz (Neurology, Digitimer, Welwyn Garden City, UK; PowerLab, ADInstruments, Colorado Springs, CO, USA). A baseline recording of 10 min was taken for basal physiological assessment before baro and chemoreceptor reflexes stimulation. Each stimulus was separated by at least 3 min.

#### 2.4.3. Baro- and Chemoreceptor Reflex Analysis

The autonomic analysis focused on assessing total autonomic tonus as well as baro- and chemoreceptor responses. The baroreceptor reflex gain (BRG) was calculated after phenylephrine provocation by measuring ΔHR/ΔBP (bpm/mmHg).

The chemoreflex response was estimated using respiratory frequency (RF) measured from tracheal pressure before and after lobeline stimulation: ΔRF = RFstimulation − RFbasal.

### 2.5. Immunohistochemistry (IHC)

At the end of the above-mentioned acute experiment, the animals were sacrificed with an overdose of the same anaesthetic compound as during the surgery (sodium pentobarbital 20% (*v*/*v*) of the 60 mg/kg initial solution).

After sacrifice, animals underwent a transcardial perfusion with fresh PBS 1× solution (TH Geyer, Höxter, Germany); the brain was removed and transferred into paraformaldehyde (PFA 4%, Carl Roth, Karlsruhe, Germany) solution for post-fixation for 72 h. The brains were then immersed in increasing concentrations of sucrose (15% and 30%, Merck, Germany in PBS 1× with sodium azide, Sigma-Aldrich, Schnelldorf, Germany) and kept at 4 °C for later analysis.

To evaluate the astrocytic changes, sagittal free-floating sections (30 μm) were cut around the region of the hippocampus (Lateral 0.6–2.04) using a cryostat (Leica CM 3050S, Wetzlar, Germany), and they were subsequently stained as reported previously [62,64]. Succinctly, free-floating sections were washed with TBS 1× (PanReac/AppliChem, Darmstadt, Germany), permeabilized with 3% Triton × 100 solution for 15 min (Carl Roth, Karlsruhe, Germany) and blocked with 5% Goat Serum (BioWest, Nuaillé, France) and 1% Bovine serum (VWR, Radnor, PA, USA) for 1 h. The freely floating tissues were then incubated with specific astrocytic primary antibody, glial fibrillary acidic protein–GFAP (chicken, ab4674, abcam, 1:500) overnight at 4 °C. Sections were then washed with TBS 1× and incubated with Alexa Flour 633 goat anti-chicken (A21103, Invitrogen, Carlsbad, CA, USA 1:1000) secondary antibody. Lastly, nuclei were counter-stained with DAPI (4′,6-diamidino-2-phenylindole, 1:10,000, Carl Roth, Karlsruhe, Germany). Sections were mounted in SuperFrost^®^ Microscope Slides using Fluoromount-G mounting media (Invitrogen, Carlsbad, CA, USA). Omission of the primary antibody resulted in no staining.

Z-stack images of the dentate gyrus region of the hippocampus were taken with a confocal point-scanning microscope (Zeiss LSM 900 with Airyscan 2, Carl Zeiss AG, Oberkochen, Germany) with a 20× objective (Objective Plan-Apochromat 20×/0.8) and tile scan was performed to image the whole region of interest and 63× objective (Objective Plan-Apochromat 63×/1.4 Oil DIC M27) for zoom in images of the cells for morphological evaluation. Zen Microscopy Software (3.4 version, Carl Zeiss AG, Oberkochen, Germany) was used for all the imaging experiments, and the final images were post-analysed and quantified using Fiji open-source software, Java V 1.8.0_322 [65].

For the morphological evaluation of the cells, relevant scientific articles were used for support [66,67,68]. For quantification of the percentage of GFAP-positive cells, the dentate gyrus region of the 20× tile images were selected, and a 3D object counter plugin in Fiji was used to first count the DAPI-stained nuclei and afterwards count the number of GFAP-positive cells. The following equation was applied for calculating the percentage of cells: (GFAP-positive cells/DAPI nuclei) × 100. Fluorescence intensity inside the cells and their area were measured using measure analysis in Fiji for the 63× images with cells individually selected for analysis.

### 2.6. Statistical Analysis

If not otherwise specified, the data were presented as mean ± SEM and represented the average of mean values across all participants. The normality distribution of continuous variables was assessed using the D’Agostino and Pearson normality test, and the homogeneity of variance was examined using Levene’s test. For data analysis among different groups, a one-way ANOVA was employed, followed by Tukey’s multiple comparisons test. We evaluated the data of both intergroups for the permanent and intermittent groups compared to controls and inter-gender groups, within each type of lead exposure comparing the male and female groups.

Additionally, to compare the exploration time percentage between objects in the NOR test for each group, a Student *t*-test for paired observations was used. The statistical analysis was performed using GraphPad Prism 9 (GraphPad Software Inc., V 9, Boston, MA, USA). Statistical significance was defined as *p* < 0.05.

## 3. Results

### 3.1. Male Lead Exposure Groups Have an Increase in Their Weight

Atomic absorption spectrophotometry was performed to estimate blood lead levels, and we observed that, regardless of gender, the IntPb and PerPb groups showed a significant increase in the levels of lead in the blood after the experimental protocol (Table 1).

As for the metabolic parameters evaluated using the metabolic cages for 24 h, we observed that male animals, both IntPb and PerPb, showed a significant decrease in their weight when compared to the controls, a change that was not observed in the female animals in both groups. We also observed a significant decrease in urine production in the male IntPb animals, compared to controls. No significant changes in the other parameters (water and food intake and faeces production) were observed (Table 1).

### 3.2. Male Permanent Exposure Group Shows a Significant Decrease in the Travelled Distance

The open-field test was used to evaluate their locomotor skills and exploratory activities. We observed that the male PerPb group of animals showed a significant decrease in their total travelled distance in the maze when compared to the male control group and compared to the female PerPb group (Figure 1A: Ctrl M 1720 ± 202.9 vs. PerPb M 1071 ± 124.4, *p* < 0.05; PerPb F 1976 ± 270.2 vs. PerPb M, *p* < 0.05). No other significant differences were observed for this parameter (Figure 1A: Ctrl F 2196 ± 166.7 vs. IntPb F 2130 ± 235.1 vs. PerPb F; Ctrl M vs. IntPb M 1188 ± 191.6 vs. PerPb M, *p* > 0.05).

Interestingly, even though the PerPb group showed a decrease in the total travelled distance, no significant differences were observed in the average velocity between all groups (Figure 1B: Ctrl F 9.5 ± 1.5 vs. IntPb F 10.1 ± 1.4 vs. PerPb F 9.1 ± 1.7 vs. Ctrl M 9.3 ± 1.9 vs. IntPb M 5.8 ± 1.2 vs. PerPb M 5.7 ± 1.2, *p* > 0.05).

### 3.3. Permanent Lead Exposure, Independent of Gender Causes Anxiety-like Behavior

Anxiety-like behaviour was evaluated using an elevated plus maze test, and the results are presented in Figure 2. We observed that the PerPb group, regardless of gender, showed a significant decrease in their percentage of time in the open arms (Figure 2A: Ctrl F 23.82 ± 4.83 vs. PerPb F 4.22 ± 1.02, *p* < 0.01; Ctrl M 17.61 ± 4.12 vs. PerPb 2.97 ± 1.48, *p* < 0.05). Remarkably, the IntPb group did not show the same effect, exhibiting a slight, not significant, decrease (Figure 2A: Ctrl F vs. IntPb 15.58 ± 3.83, *p* > 0.05; Ctrl M vs. IntPb M 9.13 ± 2.28, *p* > 0.05). No significant differences were also observed within the groups when genders were compared.

As for the percentage of time spent in the closed arms, a similar change was observed in the PerPb group, with an increase in this parameter, regardless of the gender (Figure 2B: Ctrl F 53.28 ± 5.24 vs. PerPb F 81.41 ± 2.84, *p* < 0.01; Ctrl M 53.39 ± 5.26 vs. PerPb M 80.62 ± 4.11, *p* < 0.01). The IntPb group, as with open arms, showed a slight increase, although it was not significant (Figure 2B: Ctrl F vs. IntPb F 61.85 ± 4.52, *p* > 0.05; Ctrl M vs. IntPb M 67.19 ± 5.77, *p* > 0.05). No other significant differences were observed between genders inside the groups (Int and PerPb).

### 3.4. Female Intermittent Lead Exposure and Male Permanent Lead Exposure Groups Show a Strong Decline in Episodic Long-Term Memory

Episodic long-term memory was assessed using a novel object recognition test with a 24 h retention interval and the exploration time of the novel (N) and familiar (F). We observed that the control groups, regardless of gender, recognize the novel object as novelty by showing a significant increase in the percentage of exploration time of the novel object, when compared to the familiar object (Figure 3A: Ctrl F (F) 42.28 ± 2.51 vs. Ctrl F (N) 57.72 ± 2.51, *p* < 0.01; Ctrl M (F) 39.23 ± 2.94 vs. Ctrl M (N) 60.77 ± 2.94, *p* < 0.01). Interestingly, all the other groups, irrespective of the gender or type of lead exposure, did not present a significant change in the exploration time percentage (Figure 3A: IntPb F (F) 54.78 ± 4.12 vs. IntPb F (N) 45.22 ± 4.12; IntPb M (F) 45.32 ± 2.35 vs. IntPb M (N) 54.68 ± 2.35; PerPb F (F) 47.45 ± 2.07 vs. PerPb F (N) 52.55 ± 2.07; PerPb M (F) 53.58 ± 2.30 vs. PerPb M (N) 46.43 ± 2.30, *p* > 0.05).

Remarkably, even though we observed a similar pattern in the percentage of exploration time in the different lead-exposed groups, after calculating the novel object recognition index, we observed not only different lead exposure profile effects, but also gender effects within the groups. Namely, only the IntPb females and PerPb males showed a significant decrease in the novel object recognition index, when compared to controls (Figure 3B: Ctrl F 0.15 ± 0.05 vs. IntPb F −0.13 ± 0.08, *p* < 0.01; Ctrl M 0.24 ± 0.06 vs. PerPb M −0.11 ± 0.04, *p* < 0.001), while the other two groups—IntPb M and PerPb F—showed a slight, not significant, decrease in this parameter (Figure 3B: Ctrl F vs. PerPb F 0.08 ± 0.05; Ctrl M vs. IntPb 0.09 ± 0.05, *p* > 0.05). Due to these variations, we observed a significant difference between the genders of the two lead exposure groups (Figure 3B—IntPb F vs. IntPb M, *p* < 0.05; PerPb F vs. PerPb M, *p* < 0.01).

### 3.5. Lead Exposure, Independent of Gender, Causes Astrocytic Activation with Stronger Effects in the Male Groups

In order to evaluate the molecular changes underlying the behavioural changes that were observed, we have evaluated the astrocytic cells. The qualitative morphological analysis showed that all groups, regardless of the type of lead exposure or gender, experience a significant increase in the activation of the astrocytic cells. The representative images show a higher density in the astrocytic cells, with hypertrophy of cellular processes and GFAP upregulation, which qualitatively shows that the astrocytes are in the reactive state (hallmark of pathology—Figure 4A).

As for the quantitative analysis, we observed that, regardless of the gender or type of lead exposure, the percentage of astrocytic cells increased when compared to control groups (Figure 4B: Ctrl F 11.33 ± 2.36 vs. IntPb F 40.17 ± 1.87, *p* < 0.001; Ctrl F vs. PerPb F 35.75 ± 2.53, *p* < 0.01; Ctrl M 13.28 ± 3.11, *p* < 0.001; Ctrl M vs. PerPb M 45.12 ± 7.41, *p* < 0.01). Remarkably, no changes within the groups regarding gender differences were observed (*p* > 0.05). The quantification of the GFAP fluorescence intensity showed that only the IntPb M group display a significant increase in the fluorescence intensity (Figure 4C: Ctrl M 46.84 ± 8.90 vs. IntPb M 111.8 ± 14.05, *p* < 0.01). All the other groups showed no significant changes (Figure 4C: Ctrl F 60.02 ± 4.23 vs. IntPb F 74.12 ± 2.49 vs. PerPb F 66.05 ± 12.32, *p* > 0.05; Ctrl M vs. PerPb M 58.14 ± 6.31, *p* > 0.06). No significant alterations within the lead exposure groups were depicted.

Lastly, the GFAP cell area was evaluated. We observed that, no matter the type of lead exposure or the gender, there is a significant increase in the area (Figure 4D: Ctrl F 406.7 ± 69.69 vs. IntPb F 968.7 ± 146.5, *p* < 0.05; Ctrl F vs. PerPb F 799.7 ± 48.96, *p* < 0.05; Ctrl M 337.3 ± 32.34 vs. IntPb M 1648 ± 30.0, *p* < 0.0001; Ctrl M vs. PerPb M 799.0 ± 153.0, *p* < 0.05).

### 3.6. Permanent Lead Exposure, Regardless of the Gender Shows a Stronger Effect on Blood Pressure and Respiratory Frequency

Basal physiological data were evaluated during the acute experiment. We observed that the permanent lead exposure group, regardless of the gender, caused a strong, significant increase in the systolic (Figure 5A: Ctrl F 147.8 ± 5.8 vs. PerPb F 169.2 ± 4.6, *p* < 0.01; Ctrl M 139.9 ± 12.4 vs. PerPb M 165.4 ± 7.2, *p* < 0.05) and diastolic (Figure 5A: Ctrl F 102.4 ± 3.6 vs. PerPb F 134.9 ± 4.3, *p* < 0.0001; Ctrl M 106.8 ± 9.8 vs. PerPb M 144.9 ± 4.4, *p* < 0.001) blood pressure. Consequently, the mean blood pressure was also significantly increased (Figure 5A: Ctrl F 122.4 ± 3.5 vs. PerPb F 150.9 ± 4.2, *p* < 0.0001; Ctrl M 120.7 ± 10.7 vs. PerPb M 159.3 ± 4.2, *p* < 0.001). As for the intermittent lead exposure, we observed that only diastolic blood pressure was increased significantly (Figure 5A: Ctrl F vs. IntPb F 124.2 ± 5.2, *p* < 0.01; Ctrl M vs. IntPb M 127.2 ± 4.7, *p* < 0.05) and, thus, the mean blood pressure was increased (Figure 5A: Ctrl F vs. IntPb F 140.0 ± 5.9, *p* < 0.05; Ctrl M vs. IntPb M 143.1 ± 5.1, *p* < 0.05). No significant changes were observed in the systolic blood pressure (Figure 5A: Ctrl F vs. IntPb F 158.9 ± 7.5, *p* > 0.05; Ctrl M vs. IntPb M 165.4 ± 7.2, *p* > 0.05).

We observed no significant changes in the heart rate, regardless of the lead exposure or gender (Figure 5B: Ctrl F 391.9 ± 17.94 vs. IntPb F 384.5 ± 11.60 vs. PerPb F 406.5 ± 9.28, *p* > 0.05; Ctrl M 409.4 ± 8.83 vs. IntPb M 403.9 ± 9.70 vs. PerPb M 397.1 ± 6.67, *p* > 0.05).

Regarding the respiratory frequency, the permanent lead exposure showed a significant increase in both genders (Figure 5C: Ctrl F 54.7 ± 2.5 vs. PerPb F 68.1 ± 1.9, *p* < 0.0001; Ctrl M 72.0 ± 4.1 vs. PerPb M 88.9 ± 5.1, *p* < 0.05). Only the female intermittent lead exposure showed a significant increase (Figure 5C: Ctrl F vs. IntPb F 66.4 ± 4.0, *p* < 0.05). No significant difference was observed in the male intermittent lead exposure (Figure 5C: Ctrl M vs. IntPb M 69.1 ± 2.5, *p* > 0.05). Regarding gender differences, there was a significant increase observed in the PerPb M groups compared to the PerPb M group (*p* < 0.001).

### 3.7. Lead Exposure in Female Animals, Regardless of the Type, Increases the Chemoreflex Sensitivity

The baroreceptor reflex was calculated from the variation in blood pressure and heart rate. We observed no changes in the baroreflex gain, regardless of the lead exposure protocol or gender (Figure 6A: Ctrl F 0.49 ± 0.09 vs. IntPb F 0.48 ± 0.03 vs. PerPb F 0.48 ± 0.06, *p* > 0.05; Ctrl M 0.58 ± 0.06 vs. IntPb M 0.43 ± 0.04 vs. PerPb M 0.51 ± 0.04, *p* > 0.05).

As for the chemoreflex sensitivity, we observed that both female lead exposure groups showed a significant increase when compared to the female control group (Figure 6B: Ctrl F 17.08 ± 2.28 vs. IntPb F 32.01 ± 5.08, *p* < 0.05; Ctrl F vs. PerPb F 26.56 ± 2.95, *p* < 0.05). No significant changes were observed within the male groups, even though a slight increase was depicted (Figure 6B: Ctrl M 20.66 ± 4.04 vs. IntPb M 30.03 ± 4.13 vs. PerPb M 27.06 ± 3.09, *p* > 0.05). No changes were observed between the genders, within the lead exposure groups.

## 4. Discussion

In the present study, in general, we confirmed that permanent lead exposure, regardless of the gender, causes stronger health effects in animals when compared to the intermittent lead exposure, an effect that was already observed previously in other studies [32,48,62,69]. Likewise, regardless of the lead exposure, the blood lead levels are significantly increased and to levels that are far from the safe lead levels [1,2,14]. Remarkably, nonetheless, both lead exposure profiles, regardless of gender, show not only an increase in the percentage of astrocytes but also increase in the area of the cells which shows an astrocytic activation in the brain, namely in the dentate gyrus of the hippocampus [66,67,70]. Interestingly, intermittent lead exposure leads to stronger molecular effects, as the GFAP-stained astrocytes showed a stronger increase in the fluorescence marker which is associated with a stronger activation of these cells [67,68,70]. This effect was also specific to male animals of the group which could be related to the gender differences in the response to the toxicants and the development of astrocytes [39,71]. The more prominent effect of intermittent lead exposure could be related to a more acute response to the second lead exposure that happens in the adult stage of the animals, while in the permanent lead exposure, the CNS is constantly affected by lead and some neuronal and glial modulations have occurred through time [62].

Curiously, we observed that male animals, regardless of the type of exposure, present stronger behavioural and metabolic effects when compared to female animals. Specifically, male animals show significant weight loss with lead exposure (both intermittent and permanent) even though the animals do not reduce their daily food consumption. Weight loss has been already described as one of the effects of lead poisoning, especially in children [8,11,12,13,14,72]. Weight loss could be a confounding factor for some of the behavioural changes that we observed, namely the reduction in the total travelled distance in the open-field test that evaluated the locomotor activity of the animals. However, as the weight was not reduced from the normal values of adult rats, we can infer that the locomotor activity was mostly affected by the exposure to lead [73,74,75,76,77,78]. Interestingly, intermittent male animals also presented a reduction in urine production, without a decrease in water intake, which could mean that these animals show some renal disfunction which is one of the main health effects that has been described previously and could be one of the underlying mechanisms of high blood pressure [15,16].

Hypertension was observed in all groups of animals, no matter which gender or type of exposure, without heart rate increase. However, the PerPb groups showed a strong increase in their systolic, diastolic, and consequently mean blood pressures, which is suggestive that this exposure has a stronger effect on the cardiac output when compared to IntPb groups that show an effect only in the diastolic blood pressure, which is more affected by the peripheral vascular resistance. Hypertension has been extensively documented in the scientific literature as one of the main health effects of lead exposure [23,24,79,80,81]. However, in this study, compared to our previous results, we do not observe a concomitant baroreceptor reflex impairment, regardless of the type of exposure or gender [21,32,48,62,82,83,84]. This lack of effect strikes a great interest, as the baroreceptor reflex is one of the main mechanisms regulating blood pressure, controlled by the higher autonomic brain regions, namely PVN-NTS axis and has been greatly characterized as one of the main autonomic effects in the presence of lead [81,85,86,87,88,89]. Remarkably, only female animals showed a chemoreceptor reflex sensitivity increase, independent of lead exposure which can be related to the general alert-like reaction and could be causal of the hypertension observed in these animals. Furthermore, this increase is also suggestive that it is of great significance to maintaining oxygen homeostasis and is an important sympatoexcitatory protective mechanism as an internal defence mechanism in lead exposure. Thus, however, we do not observe changes in the baroreceptor reflexes in the animals; the autonomic function is impaired with a chemoreflex sensitivity increase, especially in the female animals. According to the literature, there are significant gender differences in the autonomic function related to the difference in the gonadal hormone presence in the central autonomic areas (oestrogen increased in females and testosterone in males) [21,35,90,91,92].

Regarding the respiratory frequency, the permanent lead exposure showed a more significant increase, and only intermittent female animals showed the same effect. Furthermore, this parameter was significantly increased in the male PerPb group when compared to their female counterparts. It is very curious to see that the gender effect differs between the different lead exposures. Thus, the separation of gender reveals that this has an overall effect in the groups, reacting differently to the lead exposures, as when we compare the groups without considering gender the effect is omitted [62]. Previous studies showed that females are more resistant to toxicants; however, chronic exposure is usually described, compared to new exposure of the intermittent nature [35,36].

Similarly, permanent lead exposure shows an anxiety-like behaviour, unrelated to the gender as shown by the results in the elevated plus maze. The reduction in time in the open arms of the maze and, consequently, the decrease in the time spent in the closed arms are significant for this behaviour. This alteration can be concomitant with the change in blood pressure, respiratory frequency, and chemoreflex hypersensitivity which could affect the animal’s behaviour and their reactivity to new, unexplored environments. This effect is well documented in the literature, not only in preclinical models but also in human studies, especially in children in which lead has been correlated to attention-deficit/hyperactivity disorder (ADHD) [25,26,29,31,62]. Cognitive impairment is another effect that has been described, and this study is not an exception as all groups of lead exposure show no recognition of the novel object in the novel object recognition test when the percentage of the exploration time is calculated (similar time exploring the familiar and the novel object or exploring the familiar object more than novel). Yet, when the novelty recognition index is calculated, the results show a dissimilar pattern for different lead exposures and genders. Solely, female IntPb and male PerPb groups show a strong NOR index reduction (below 0), an effect that was omitted when all animals were in the same group and compared between environmental lead exposure profiles. We can infer that gender is of great importance to the cognitive impairment effect, especially the hormonal differences (oestrogen levels in females and testosterone in males) that have been shown to increase neurotoxicity once following an insult but are protective in the later stages [42]. Some other possible mechanisms underlying these great varieties in the gender effect in cognition could also be the amount of microglia that males and females have a priory (males have more microglia which could result in higher neurotoxicity); the mitochondria from the female brain that have greater functional capacity, for example, for antioxidant and respiratory function; the higher mitochondrial reactive oxygen species (ROS) production in males, leading to higher susceptibility of males to neurotoxicity; epigenetics which affect the sex-specific transcriptome; lower susceptibility of females to develop glutamate-induced excitotoxicity; and other factors [34,37,42,47,93]. Interestingly, even though, according to the literature, females seem to have stronger resistance to toxicants, in our study we observe that intermittent exposure is more detrimental in females regarding cognitive performance which could be related to the second exposure that activates the neuroinflammatory and neuroprotection pathways, as well as the autonomic and physiological changes that were stronger in females in this group [30,42,94].

Our study presents some important limitations that should be considered when further studies are performed. Namely, we did not take into account the menstrual cycle of the female animals, which is an important variable that could lead to some biases and more variability in the results. We also see the importance of performing an evaluation of the physiological parameters with telemetric sensors to have the parameters evaluated in a conscious animal in natural conditions, opposite to the anaesthetized animals. Furthermore, we advocate for additional studies to delve deeper into the molecular effects of lead exposure. This entails examining not only astrocytic changes but also other markers such as microglial alterations and reactive oxygen species (ROS) production. These assessments should extend beyond the hippocampal region to encompass other relevant areas like the paraventricular nucleus of the hypothalamus (PVN), nucleus tractus solitarius (NTS), rostroventrolateral medulla (RVLN) for autonomic alterations, and the amygdala for investigating connections with anxiety-like behaviour. Moreover, studies in populations with lead exposure, especially with the new, intermittent paradigm, can bring more information regarding not only the direct effects of lead but also its cumulative effect with other factors as well as its potential for being a risk factor for various diseases.

## 5. Conclusions

In general, we show that both types of exposure to lead and gender are relevant to the detrimental health effects of lead toxicity. The permanent exposure shows that, regardless of sex, this exposure has a stronger detrimental health effect when compared to the intermittent lead exposure group and causes hypertension, anxiety, and reactive astrogliosis. However, males are more susceptible to the cognitive, behavioural, and respiratory changes, while females are more prone to chemoreflex hypersensitivity. Remarkably, intermittent lead exposure causes hypertension (diastolic blood pressure is more affected) and reactive astrogliosis, independent of gender. Conversely, female animals show higher vulnerability to cognitive impairment, respiratory frequency increase, and chemoreflex hypersensitivity, while males show more reactive astrocytes in the hippocampus.

Our results show the significance of not only studying the different types of lead exposure and their effects on health but also considering the gender differences in toxicity. By incorporating gender-specific considerations into toxicity assessments, healthcare professionals can tailor interventions to address individual needs more effectively. Similarly, policymakers can design targeted public health initiatives that account for the unique vulnerabilities of both genders, thereby optimizing resource allocation and mitigating the adverse impacts of lead exposure on public health.

Ultimately, a comprehensive understanding of gender differences in lead toxicity not only enhances the precision of treatment approaches but also strengthens the foundation for evidence-based policy interventions aimed at safeguarding the health and well-being of populations at risk.

## Figures and Tables

**Figure 1 biomedicines-12-00711-f001:**
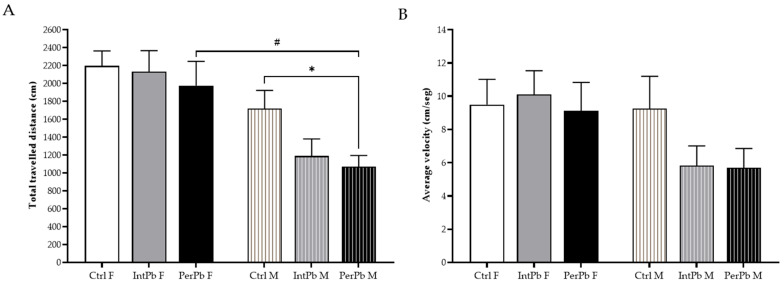
Locomotor and exploratory behaviours evaluated using the open-field test. (**A**) Total travelled distance of the animals. (**B**) Average velocity. Values are the mean ± SEM for n_Ctrl F_ = 16, n_IntPb F_ = 17, n_PerPb F_ = 17, n_Ctrl M_ = 19, n_IntPb M_ = 17 and n_PerPb M_ = 16. The symbols denote statistically significant differences intergroup (Ctrl vs. IntPb vs. PerPb—* *p* < 0.05; and inter-gender (IntPb F vs. IntPb M/PerPb F vs. PerPb M—# *p* < 0.05; one-way ANOVA; Tuckey’s multiple comparison test.

**Figure 2 biomedicines-12-00711-f002:**
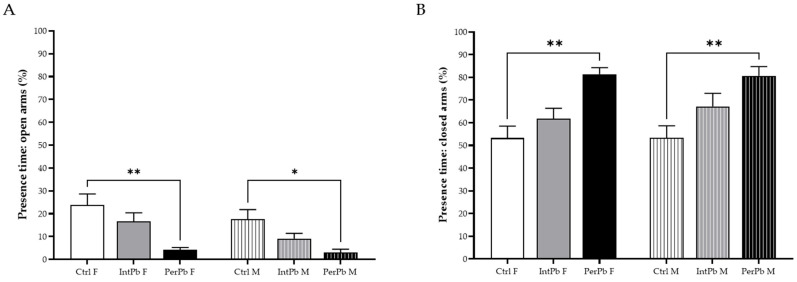
Anxiety-like behaviour results assessed using elevated plus maze. (**A**) Percentage of time in open arms. (**B**) Percentage of time in closed arms. Values are the mean ± SEM for n_Ctrl F_ = 10, n_IntPb F_ = 15, n_PerPb F_ = 13, n_Ctrl M_ = 16, n_IntPb M_ = 13, and n_PerPb M_ = 12. The symbols denote statistically significant differences intergroup (Ctrl vs. IntPb vs. PerPb—* *p* < 0.05; ** *p* < 0.01; one-way ANOVA; Tuckey’s multiple comparison test.

**Figure 3 biomedicines-12-00711-f003:**
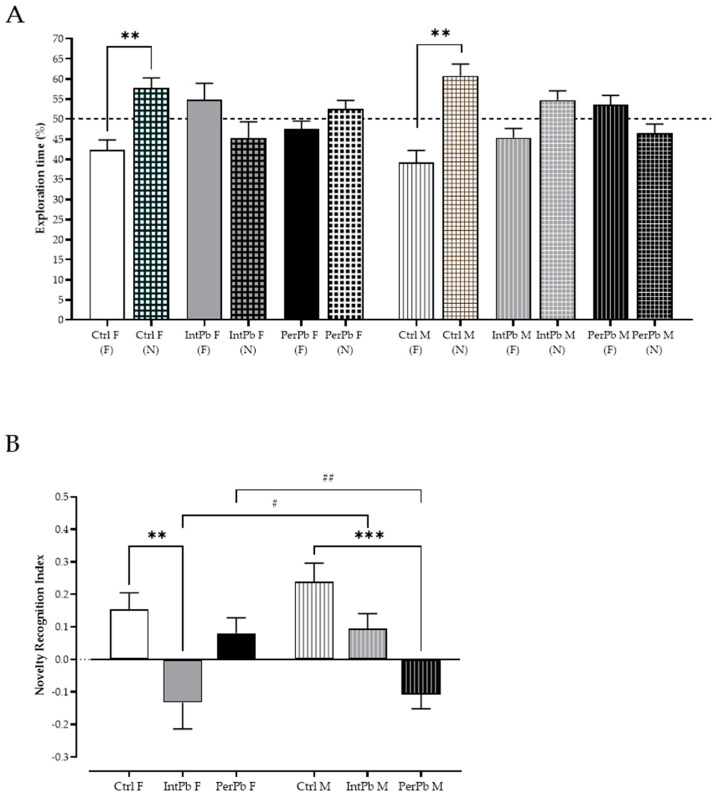
Episodic long-term memory assessed using novel object recognition test. (**A**) Exploration time percentage of familiar (F) and novel (N) objects by each group. (**B**). Novel object recognition index calculated through the equation presented in the methods section. Dotted line represents the 50% exploration time %. Values are the mean ± SEM for n_Ctrl F_ = 14, n_IntPb F_ = 11, n_PerPb F_ = 12, n_Ctrl M_ = 17, n_IntPb M_ = 15, and n_PerPb M_ = 14. The symbols denote statistically significant differences: (**A**) Between the exploration time of the novel (N) and familiar (F) objects (** *p* < 0.01). Paired Student’s *t*-test. (**B**) Intergroup (Ctrl vs. IntPb vs. PerPb—** *p* < 0.01; *** *p* < 0.001) and inter-gender (IntPb F vs. IntPb M/PerPb F vs. PerPb M—# *p* < 0.05; ## *p* < 0.01). One-way ANOVA. Tuckey’s multiple comparison test.

**Figure 4 biomedicines-12-00711-f004:**
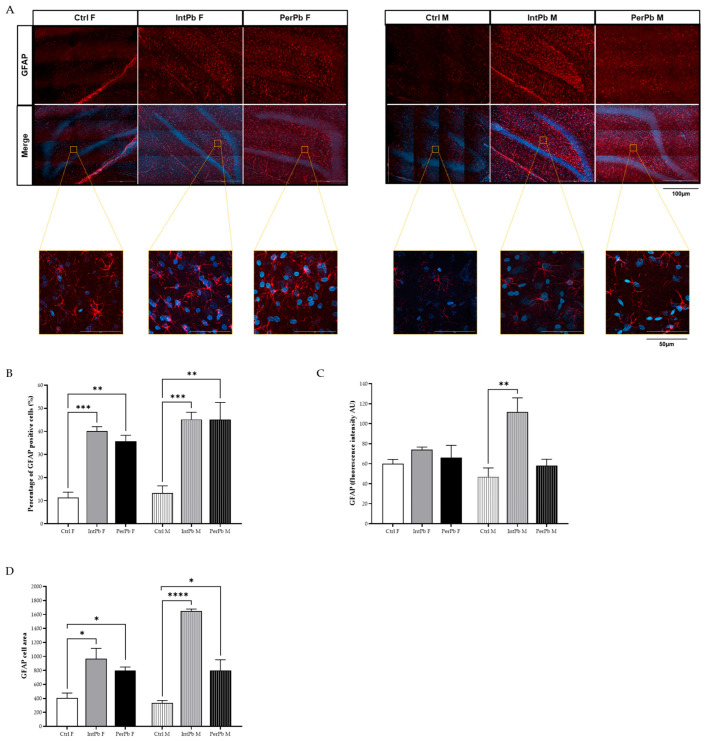
Astrocytic activation evaluated using astrocytic (GFAP) marker using immunohistochemistry. (**A**) Representative images of the GFAP-stained astrocytes. (**B**) Histogram of the percentage of GFAP-positive cells’ quantification. (**C**). Histogram of GFAP fluorescence intensity quantification. (**D**) Histogram of GFAP cell area quantification. Images were acquired on a confocal point scanning microscope, (Zeiss LSM 900 with Airyscan 2), with a 20× objective for a whole dentate gyrus imaging and 63× objective for zooming in on the astrocytic cells. The scale bar is 100 µm or 50 µm for stained images. Values are the mean ± SEM for *n* = 2–4 for all groups. The symbols denote statistically significant differences: intergroup (Ctrl vs. IntPb vs. PerPb—* *p* < 0.05; ** *p* < 0.01; *** *p* < 0.001; **** *p* < 0.0001); one-way ANOVA; Tuckey’s multiple comparison test.

**Figure 5 biomedicines-12-00711-f005:**
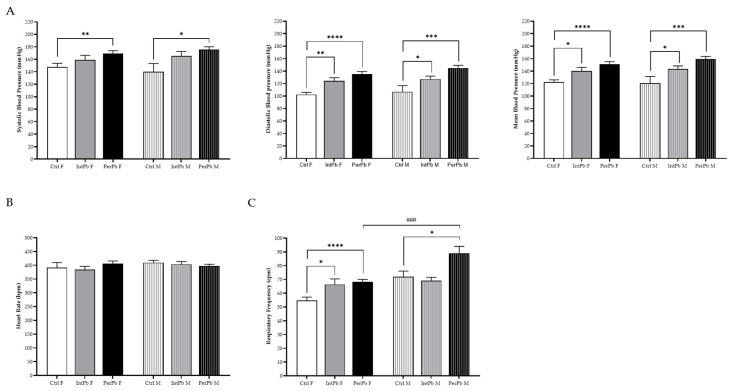
Basal physiological data assessed during the acute surgery. (**A**) Systolic, diastolic, and mean blood pressure evaluated through femoral artery. (**B**) Heart rate calculated from the electrocardiogram. (**C**) Respiratory frequency assessed from tracheal pressure. Values are the mean ± SEM for n_Ctrl F_ = 10, n_IntPb F_ = 16, n_PerPb F_ = 15, n_Ctrl M_ = 9, n_IntPb M_ = 15, and n_PerPb M_ = 16. The symbols denote statistically significant differences intergroup (Ctrl vs. IntPb vs. PerPb—* *p* < 0.05; ** *p* < 0.01; *** *p* < 0.001; **** *p* < 0.0001) and inter-gender (IntPb F vs. IntPb M/PerPb F vs. PerPb M—### *p* < 0.001). One-way ANOVA. Tuckey’s multiple comparison test.

**Figure 6 biomedicines-12-00711-f006:**
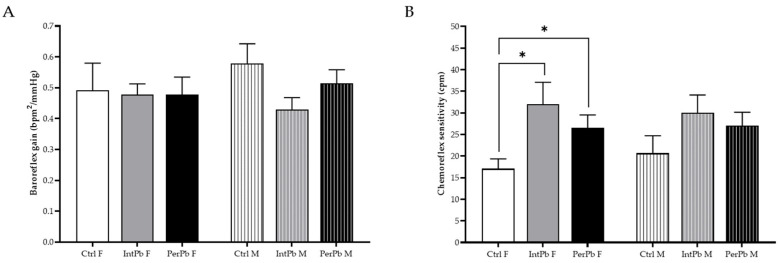
Lead effect on autonomic reflexes. (**A**) Baroreflex gain stimulated with phenylephrine injection. (**B**) Chemoreflex sensitivity calculated from lobeline injection stimulation. Values are the mean ± SEM for n_Ctrl F_ = 10, n_IntPb F_ = 12, n_PerPb F_ = 15, n_Ctrl M_ = 9, n_IntPb M_ = 12 and n_PerPb M_ = 16. The symbols denote statistically significant differences intergroup (Ctrl vs. IntPb vs. PerPb—* *p* < 0.05). One-way ANOVA. Tuckey’s multiple comparison test.

**Table 1 biomedicines-12-00711-t001:** Gender-specific changes in blood lead levels and metabolic parameters. Values are the mean ± SEM. The symbols denote statistically significant differences intergroup (Ctrl vs. IntPb vs. PerPb—* *p* < 0.05; ** *p* < 0.01; *** *p* < 0.001; **** *p* < 0.0001); one-way ANOVA; Tuckey’s multiple comparison test.

	Blood Lead Levels (μg/dL)	Weight (g)	Food Intake (g)	Water Intake (mL)	Produced Faeces (g)	Produced Urine (mL)
Ctrl F	0.425 ± 0.085	329 ± 8	24.9 ± 1.3	40.0 ± 2.4	28.30 ± 3.07	16.31 ± 1.77
IntPb F	17.85 ± 5.350 *	302 ± 6	23.6 ± 1.7	45.9 ± 6.1	21.00 ± 4.25	16.36 ± 2.23
PerPb F	26.43 ± 3.798 ***	324 ± 7	23.2 ± 1.9	37.3 ± 4.6	21.31 ± 3.62	16.00 ± 1.99
Ctrl M	0.48 ± 0.086	620 ± 15	36.3 ± 3.4	47.1 ± 2.6	29.14 ± 4.96	21.00 ± 2.38
IntPb M	18.77 ± 0.612 **	501 ± 7 ****	31.5 ± 4.6	37.5 ± 5.2	24.33 ± 3.44	12.83 ± 0.58 **
PerPb M	21.77 ± 5.871 ***	555 ± 17 **	29.7 ± 1.3	34.3 ± 3.6	21.43 ± 3.96	17.00 ± 1.43

## Data Availability

The data presented in this study are available on request from the corresponding author. The data are not publicly available due to ethical restrictions.

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
