# Peer review of "Gender-Specific Effects on the Cardiorespiratory System and Neurotoxicity of Intermittent and Permanent Low-Level Lead Exposures"

_biomedicines, 2024, doi:10.3390/biomedicines12040711_

Round 1
Reviewer 1 Report
Comments and Suggestions for Authors
The manuscript addresses a significant and underexplored area of public health research. The focus on gender differences in response to lead exposure is timely and relevant. Recommendations for improvement include providing more detailed methodological information, enhancing the statistical analysis section, and further discussing the results in the context of existing literature and potential physiological mechanisms. The potential impact of this research on public health policy and preventive strategies should be highlighted, along with the necessity for human studies to confirm these findings and translate them into clinical practice.
Carefully proofread the manuscript to correct any typographical or grammatical errors, as this will improve the overall readability and professionalism of the text. Consider the inclusion of a supplementary section or appendix for any extensive datasets or methodological details that could aid in replicating your study but would clutter the main text. These comments are aimed at refining the manuscript to meet the high standards of academic publishing, ensuring that your valuable research is communicated effectively and contributes meaningfully to the field.
Recommend for publication with minor revisions, focusing on methodological details, statistical analysis clarity, and expanded discussion of implications and future directions.
Author Response
Thank you for the remarks and the suggestions for improving the manuscript.We have revised the whole manuscript and made the suggested chnages along the whole document.
Reviewer 2 Report
Comments and Suggestions for Authors
This is a well written and apparently well-designed and performed study looking at the effect of oral Pb on both male and female rat pups. Pb was given so as to model intermittent and continuous lead exposure with neurological and other toxicological markers being analyzed. The results will be notable to other in the field.
After carefully reading the manuscript, I have only a single minor issue:
I did not find the euthanasia procedure.
Author Response
Thank you for the comments and remarks. Indeed, the euthanasia procedure is not fully explained in the methods, refering only to the overdose of the anaesthesia that was used during the surgical procedure. We have reviewed the mansucript and made it more clear in the methods what is the method of euthanasia. We added the following information: “At the end of the above-mentioned acute experiment, the animal was sacrificed with an overdose of the same anaesthetic compound as during the surgery (sodium pentobarbital 20% (v/v) of the 60 mg/kg initial solution).”